# Study protocol on risk factors for the diagnosis of gestational diabetes mellitus in different trimesters and their relation to maternal and neonatal outcomes (GDM-RIDMAN)

Pamela Phui Har Yap ,[1] Iliatha Papachristou Nadal,[2,3] Veronika Rysinova,[3] Nurul Iftida Basri,[4] Intan Nureslyna Samsudin,[5] Angus Forbes ,[6] Nurain Mohd Noor,[7] Ziti Akthar Supian,[8] Haslinda Hassan,[9] Fuziah Paimin,[10] Rozita Zakaria,[11] Siti Rohani Mohamed Alias,[12] Norizzati Bukhary Ismail Bukhary,[13] Madeleine Benton,[3] Khalida Ismail,[3] Boon How Chew [1,3,14]

For numbered affiliations see end of article.

**Correspondence to**
Dr Boon How Chew;
chewboonhow@upm.edu.my

## ABSTRACT

**Introduction** Gestational diabetes mellitus (GDM) is often associated with adverse pregnancy outcomes. However, the association of risk factors with GDM diagnosis, maternal and neonatal health outcomes is less established when compared with women without GDM. We aim to examine the diagnostic accuracy of the conventional and novel risk factors for a GDM diagnosis and their impact on maternal and neonatal health outcomes.

**Methods and analysis** This retrospective cohort and nested case–control study at six public health clinics is based on medical records and questionnaire survey of women between 2 and 12 months postpartum. The estimated required sample size is 876 complete records (292 cases, 584 control, at a ratio of 1:2). Oral glucose tolerance test results will be used to identify glucose dysregulation, and maternal and neonatal outcomes include maternal weight gain, pre-eclampsia, polyhydramnios, mode of delivery, preterm or postdate birth, complications in labour, birth weight, gestational age at birth, Apgar score, congenital anomaly, congenital hypothyroidism, neonatal death or stillbirth, hypoglycaemia and hyperbilirubinaemia. Psychosocial measures include the WHO Quality of Life: brief, mother–infant bonding (14-item Postpartum Bonding Questionnaire and 19-item Maternal Postnatal Attachment Scale), anxiety (7-item Generalised Anxiety Disorder), depression (9-item Patient Health Questionnaire) and stress (Perceived Stress Scale symptoms) questionnaires. The comparative incidences of maternal and neonatal health outcomes, the comparative prevalence of the psychosocial outcomes between women with GDM and without GDM, specificity, sensitivity, positive and negative predictive values of the risk factors, separately and combined, will be reported. All GDM risk factors and outcomes will be modelled using multivariable regression analysis and the receiver operating characteristics curve will be reported.

**Ethics and dissemination** This study was approved by the Malaysia Research and Ethics Committee, Ministry of Health Malaysia. Informed consent will be obtained from all participants. Findings will be submitted for publications in scientific journals.

## Strengths and limitations of this study

⇒ This study provides an opportunity to confirm and explore the conventional risk factors to better predict the diagnosis of gestational diabetes mellitus.
⇒ Quality of life, mental health and maternal–infant bonding will be assessed during the COVID-19 pandemic.
⇒ Participants are mainly from the urban areas of Selangor and Putrajaya and may not be representative of a larger population.
⇒ The incidence of some of the risk factors may be low and insufficient for inclusion into multivariable regression analysis.
⇒ Recruitment of fewer participants than expected due to the COVID-19 pandemic and the reduced number of pregnancies and postpartum women visiting clinics in-person may introduce challenges in data collection, analysis and interpretation.

## INTRODUCTION

Gestational diabetes mellitus (GDM) is a form of hyperglycaemia, where pregnant women experience glucose intolerance for the first time during the pregnancy.[1–3] Efficient GDM screening and accurate diagnosis allow for early management and treatment which could reduce adverse pregnancy outcomes for both mother and child.[4–6]

Asian women are at a higher risk for GDM compared with Caucasian women.[7 8] A systematic review and meta-analysis reported that the prevalence of GDM in Malaysia is in the top five Eastern and Southeast Asian

countries, where approximately one in nine pregnant women had GDM (11.8%).[8] The latest National Obstetric Registry 2016–2017 reported the prevalence of GDM with adverse outcomes in Malaysia range from 10.8% to 19.3%.[9] Spontaneous miscarriage and caesarean section are the most frequently reported adverse outcomes in GDM women compared with healthy women, 5.9% versus 2.6% and 28.5% versus 18.8%, respectively.[10] Other adverse maternal outcomes include birth trauma, postpartum haemorrhage, pre-eclampsia and hypertension (≥140/90 mm Hg) after the 20th week of gestation with proteinuria.[10 11] Common neonatal adverse outcomes include foetal macrosomia, hypoglycaemia, prematurity, shoulder dystocia, hyperbilirubinaemia and admission to intensive care units.[11 12] Prevalence of macrosomic babies and neonatal hypoglycaemia among GDM mothers in Malaysia in 2018 are 4.8% and 1.7%, respectively.[12] Postdiagnosis postprandial glucose levels are associated with macrosomia and large for gestational age (LGA), premature delivery, gestational hypertension and hyperbilirubinaemia.[13–15]

At both the local and international levels, screening methods and diagnosis of GDM remain debatable. High fasting plasma glucose (FPG) level significantly increases the risk for LGA fetus, primary caesarean section and development of GDM in later pregnancy.[16] Several studies from Israel and Asian population support FPG ≥6.10 mmol/L at first trimester as the predictor tools for GDM development in later pregnancy.[17 18] Another study in China suggested that FPG 6.1–6.9 mmol/L was more accurate and reliable at early pregnancy (before the 24th week) for GDM diagnosis. This differs from the International Association of Diabetes and Pregnancy Study Group cut-off values of FPG ≥5.1 mmol/L[18 19]. However, this is less well accepted because many perceive FPG ≥5.1 mmol/L to be a false alarm, as when pregnancy advances in a week, FPG level decreases.[18] Glycosylated haemoglobin A1c (HbA1c) is not widely used in practice as the value can be unreliable due to many confounding conditions such as hemoglobinopathy.[20] Although studies have shown the predictive value of HbA1c ≥6.0% of GDM development[5] and adverse neonatal outcomes.[10 21 22]

Malaysia practices selective risk-based screening and one-step diagnosis for GDM for early diagnosis and management. All pregnant women will be risk stratified according to the Malaysia 2017 Clinical Practice Guidelines (CPG) on Management of Diabetes in Pregnancy.[23] Women who are perceived to be at risk of GDM will undergo 75 g oral glucose tolerance test (OGTT) as soon as the next appointment between the 24th and 28th week of gestation.[23] The test requires women to stay fasted from food and drink for at least 8–12 hours for FPG and 2 hours after the oral glucose intake. The one-step diagnosis approach of GDM will diagnose GDM when at least any one single abnormal reading is observed at fasting or 2 hours from 75 g OGTT.[19 23 24] There are some challenges in full OGTT compliance, which include long hours of fasting, drinking of glucose water, vomiting and defaulting

> **Box 1    The Malaysian Clinical Practice Guideline (CPG) 2017 recommended risk factors for GDM screening**
>
> CPG risk factors:
> 1. Age ≥25 years old.
> 2. Body mass index >27 kg/m$^2$.
> 3. Previous history of Gestational diabetes mellitus.
> 4. First degree relative with diabetes mellitus.
> 5. History of macrosomia (birth weight >4 kg).
> 6. Bad obstetric history (unexplained intrauterine death, neural tube defects, cardiac defects and shoulder dystocia).
> 7. Glycosuria ≥2+ on two occasions.
> 8. Current obstetric problems (essential hypertension, pregnancy-induced hypertension (≥140/90 mm Hg), polyhydramnios and current use of corticosteroids).

the 2 hour postprandial (2-HPP) plasma glucose test.[25–27] However, there is uncertainty in terms of the state of OGTT compliance and completion rate during the first or second trimester. Similarly, there is a lack of information about the completion rate of OGTT and its association with adverse pregnancy outcomes among pregnant women before the GDM diagnosis.[11] Therefore, adherence of OGTT is equally important during and after pregnancy for GDM women.

The common risk factors of GDM include age ≥25 years old, body mass index >27 kg/m$^2$, previous history of GDM, first degree relative with diabetes mellitus, history of macrosomia (birth weight >4 kg), bad obstetric history (unexplained intrauterine death, neural tube defects, cardiac defects and shoulder dystocia), glycosuria ≥2+ on two occasions, current obstetric problems (essential hypertension, pregnancy-induced hypertension (≥140/90 mm Hg), polyhydramnios and current use of corticosteroids[23] (see box 1). Other potential risk factors for GDM include polycystic ovarian syndrome (PCOS) (OR 2.33, 95% CI 1.72 to 3.17), multiple pregnancies (OR 1.37%, 95% CI 1.24 to 1.52), preterm birth (OR 1.93, 95% CI 1.21 to 3.07),[24] maternal gestational weight gain (adjusted OR 3.38%, 95% CI 1.83 to 6.24)[28] and maternal smoking status (OR 1.22, 95% CI 1.08 to 1.38).[29] The prevalence of PCOS among Malaysian working women age 18–49 years was 12.6%,[30] but PCOS is not considered to be a risk factor in the Malaysian CPG. Multiple pregnancies is associated with higher risk for GDM due to a higher weight gain rather than the number of fetus.[31] The rate of multiple pregnancy deliveries among GDM women in Malaysia range approximately from 12% to 30%.[32 33] Many GDM studies have excluded multiple pregnancies in their eligibility criteria, so the risk of multiple pregnancies on GDM is not clear. There is conflicting evidence on smoking and GDM.[34 35]

### Psychosocial aspects in the postpartum period

There is increasing evidence demonstrating that psychosocial factors, such as depressive symptoms, anxiety, high perceived stress, poor quality of life (QoL) and weakened mother-child bonding play an important role in GDM

---

**Box 2  Potential new risk factors to be screened for Gestational diabetes mellitus diagnosis and health outcomes**

Potential risk factors:
⇒ History of polycystic ovarian syndrome.
⇒ Current multiple pregnancies.
⇒ Active or passive smoking status.
⇒ Miscarriage (before 23rd week) (*previous and most present*).
⇒ Preterm birth (23rd to 36th week +6 days) (*previous and most present*).
⇒ Gestational weight gain.

and adverse neonatal outcomes.[12 36–38] A systematic review demonstrated that GDM respondents consistently showed significantly lower QoL both short-term and long-term compared with healthy pregnant participants.[37] Although the QoL scores significantly improved after pregnancy in healthy women, general health perception in GDM women remained significantly lower.[39 40] The estimated prevalence of depression, anxiety, stress symptoms and poor QoL in GDM women in Malaysia ranges from 10.2% to 39.9%.[12]

Additionally, the odds of GDM were found to be 13-fold higher in women with high stress levels during pregnancy than in women with low stress levels among Indian women,[41] suggesting that high perceived stress is a potential risk factor for GDM development. Similarly, a recent systematic review and meta-analysis investigating 62 studies indicated that there was an increased risk of depression and anxiety symptoms around the time of GDM diagnosis

and in the postnatal period.[42] Weakened mother–child bonding was previously shown to have a statistically significant association with the mother's depression and anxiety symptoms and these variables also affected mother–child bonding.[36] Mother–child bonding is crucial to the mental growth and development of infants.[43] Given the potential link between GDM and psychosocial factors, investigation and integration of physical and mental health factors in empirical studies and interventions with women experiencing GDM in Malaysia is, therefore, vitally important and could improve short-term and long-term outcomes for women and their children.[42]

### Conceptual framework

The independent variables are risk factors of GDM, including both the CPG-based (box 1) and the potential risk factors (box 2) based on the literature. The first dependent variables are the diagnosis of GDM in the first or second trimester as recorded in the maternal records. The second separate dependent variables in this study are the maternal and neonatal health outcomes, including the maternal psychological well-being. The effect of glycaemic patterns after GDM diagnosis on the maternal and neonatal health outcomes is believed to be present and will be examined. Additionally, we aim to describe the incidences of the maternal and neonatal outcomes during pregnancy and in the postpartum period in women with GDM and without GDM. Figure 1 shows the overall conceptual framework of this study.

There are uncertainties about the differences in risk factor profiles, the antenatal hyperglycaemia patterns,

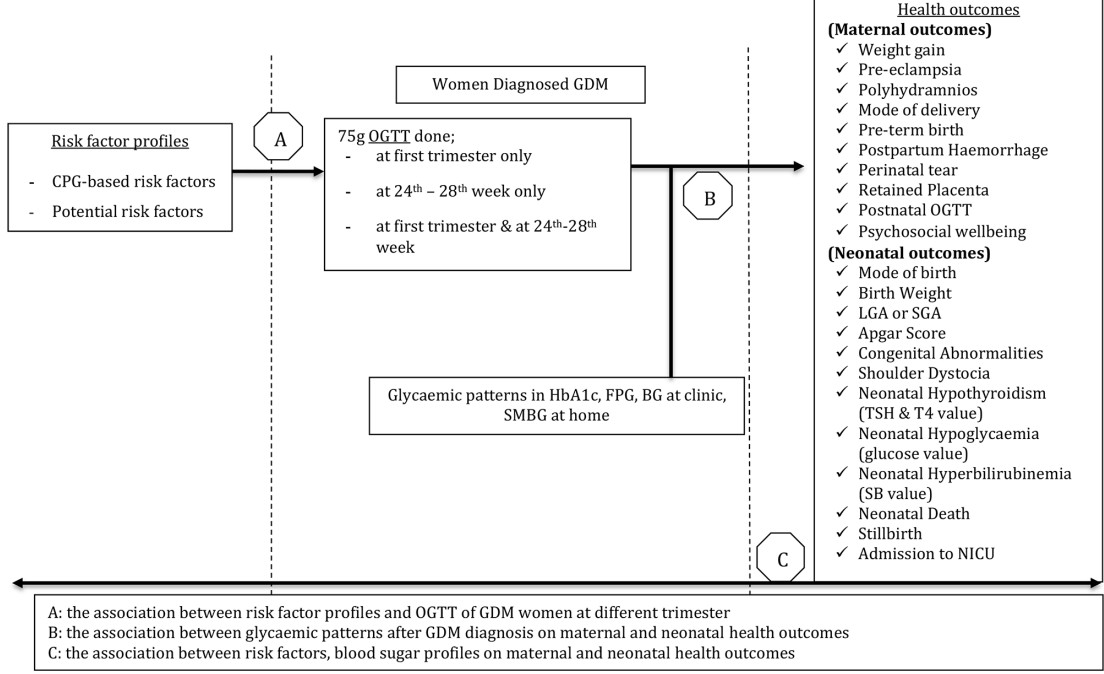

**Figure 1**  Risk factors for the diagnosis of gestational diabetes mellitus in different trimester and their relation to maternal and neonatal outcomes. *BG, blood glucose; CPG, clinical practice guidelines; FPG, fasting plasma glucose; LGA, large gestational age; NICU, neonatal intensive care unit; OGTT, oral glucose tolerance test, SB, serum bilirubin; SGA, small gestational age; SMBG, self-monitoring blood glucose; TSH, thyroid stimulating hormone; T4, thyroxine.

and also the difference on pregnancy outcomes of those who completed the OGTT in the first or second trimester with a GDM diagnosis. Therefore, this study aims to examine and quantify the effects of the current recommended CPG-based risk factors for GDM diagnosis and to compare the risk factor profiles between those associated with GDM diagnoses in the first and second trimesters. We will also examine the effects of other potential new risk factors on the actual diagnosis of GDM. Additionally, we will investigate the impacts of all the risk factors and the postdiagnosis blood sugar profiles on the maternal and neonatal outcomes.

## METHOD AND ANALYSIS

This retrospective cohort and nested case–control study will be based on antenatal and postnatal medical records and will include a questionnaire survey of postpartum women who have delivered within the past 2–12 months. The five study objectives are:

1. To compare the risk factor profiles of women who have undergone OGTT and have been diagnosed with GDM in the first and second trimesters.
2. To describe the occurrences of the maternal health outcomes during pregnancy and in the postpartum period between those with GDM and without GDM.
3. To describe the incidence of the neonatal health outcomes of mothers with GDM and without GDM.
4. To identify and quantify risk factors (including blood sugar profiles) that are associated with any maternal complications.
5. To identify and quantify risk factors (including blood sugar profiles) that are associated with any neonatal complications.

## Settings

Data will be collected from six participating public health clinics in Selangor and Putrajaya over approximately 5 months. These clinics are attended by pregnant women of different ethnicities and will provide maternal and childcare services with in-house laboratory services, and availability of GDM registry. The standard care processes in these clinics are that women will be seen by nurses and medical officers or a family medicine specialist when needed for further care. GDM women usually have a follow-up appointment every 2 weeks for blood sugar level monitoring and pregnancy progress consultation. In postpartum, the appointment ranges from 4 to 6 weeks for family planning counselling and a repeat for the OGTT.[10 44]

Preprandial and postprandial glucose tests are blood sugar profile tests used for GDM monitoring. A preprandial plasma glucose should be ≤5.3 mmol/L[23 45] and postprandial values at 1 hour and 2 hours are ≤7.8 mmol/L and ≤6.7 mmol/L to be considered as optimal, respectively.[23]

## Participants

The eligible participants are Malaysian women age ≥18 years old who have undergone OGTT during the last pregnancy, both singleton and multiple pregnancies, with a baby aged at least 2 months and above (*include preterm birth between 23 and 37 week of gestation but chronological age at least 2 months*) or who have had a miscarriage during last pregnancy, receiving most antenatal care at the participating clinics during the last pregnancy and who have returned for a postnatal follow-up at the participating clinics. Malaysian women with pre-existing type 1 or type 2 diabetes and overt diabetes will be excluded from this study.

## Instruments

GDM screening and diagnosis are based on the Malaysian 2017 CPG on Management of Diabetes in Pregnancy.[23] Pregnant women with either one of the abnormal OGTT results, FPG ≥5.1 mmol/L or 2-HPP ≥7.8 mmol/L will be diagnosed as GDM.[23] All data will be retrieved from the antenatal home-based record, and if necessary, from the baby's record book, clinic-based record, GDM registry book, healthcare electronic medical records and laboratory records that are available at each participating centre. If there is insufficient or missing data from the records, we may call the participants for clarification. All variables will be recorded in a structured case record form. All data retrieved will be labelled, stored safely in hard drive and password protected. The independent and dependent variables include the following:

1. Mother's risk factors: age ≥25 years old, at booking BMI >27 kg/m$^2$, history of GDM, first degree family with diabetes mellitus, previous baby with birth weight >4 kg, poor obstetrics medical history (unexplained intrauterine death, congenital abnormalities as such neural tube defects, cardiac defects and shoulder dystocia), glycosuria ≥2+ on two occasions, medical disorders (hypertension ≥140/90 mm Hg, polyhydramnios and on corticosteroid medication), multiple pregnancies, smoking status, miscarriage occurring before 23 weeks during the previous or most present pregnancy and history of PCOS.
2. OGTT result, all blood sugar profiles (blood glucose testing performed at the clinic or home monitoring) and HbA1c level.
3. Delivery records include:
   a. maternal outcomes (gestational weight gain, pre-eclampsia, polyhydramnios, mode of delivery, gestational age at birth either preterm or postdate birth and complications in labour, including postpartum haemorrhage and perineal tear).
   b. neonatal outcomes (birth weight, gestational weight at birth either LGA or small gestational age (SGA), Apgar score, congenital anomaly, congenital hypothyroidism from thyroid-stimulating hormone and T4 levels, neonatal death and stillbirth, hypoglycaemia from plasma glucose level and hyperbilirubinaemia from serum bilirubin level).

4. Psychosocial measures include QoL,[46] mother–infant bonding,[47–49] anxiety and depression symptoms[50 51] and perceived stress[52] (table 1).

## Cultural adaptation and validation process for questionnaires

The 14-item Postpartum Bonding Questionnaire (PBQ-14) and 19-item Maternal Postnatal Attachment Scale (MPAS-19) will be translated from English to Malay by bilingual translators. Two forward translations from English to Malay will be produced and then another bilingual translator will translate the scale back into English while being blinded to the original English version of the questionnaires.[53] The two Malay versions for the scale will be compared with the original and back-translated English version. Based on discussion and consensus by an expert committee, the most appropriate Malay version will be developed and chosen in this study. The expert committee comprises three Malay women with a history of GDM, bilingual (English and Malay) two family physicians, two psychologists and one obstetrician and gynaecologist.

Principal component analysis (PCA) will be used, using the Orthogonal rotation (Varimax) to determine the subscales. Preliminary analysis of the PCA output will be made to investigate multicollinearity, sampling adequacy using the Kaiser-Meyer-Olkin statistic and Bartlett's test of sphericity. Internal consistency will be calculated with Cronbach's alpha values for all subscales and 2-week test–retest reliability will be examined with intraclass correlation coefficients (ICC). A Cronbach's alpha value of at least 0.75 indicates good internal consistency of the questionnaire.[54] ICC of at least 0.7 is preferred for a sample size of >50 subjects to estimate the test–retest reliability.[55 56]

To check for construct validity (hypothesis-testing validity),[57] associations with GDM status, maternal outcomes (weight gain, pre-eclampsia, polyhydramnios, mode of delivery, preterm or postdate birth, postpartum haemorrhage, perineal tear, retained placenta, postnatal OGTT result), neonatal outcomes (birth weight, LGA or SGA, Apgar score, congenital abnormalities, shoulder dystocia, neonatal hypothyroidism, neonatal hypoglycaemia, neonatal hyperbilirubinaemia, admission to Neonatal Intensive Care Unit) and the other psychosocial measures (WHO Quality of Life-Bref, Generalised Anxiety Disorder-7, PHQ-9 and Perceived Stress Scale-10) will be examined. Finally, PBQ-14 and MPAS-19 will be examined against each other. We hypothesise that maternal feelings of bonding are moderately correlated ($r \geq 0.5$) and associated with maternal mood.[58 59] All statistical analyses will use SPSS V.26.0 (IBM, Chicago, Illinois) and p value >0.05 will be considered statistically significant.

## Sampling process

The study will take place at all six participating health clinics at the same time. All eligible participants will be invited to participate in the study. All women with GDM will be the case. Two control will be sampled for each case. Women in the control group were selected based on the eligibility criteria from the same six health clinics who had undergone OGTT but without a diagnosis of GDM, at about the same period of gestation at the diagnosis of GDM to the case. Women who have consented will leave their home-based health record for the researcher for a period of 1–2 months. They will self-administer the questionnaires (about 30 min to complete) online or manually in either Malay or English according to their preference and return it before they leave the clinic. A trained research assistant will be at each participating clinic to facilitate the data collection and to respond to any queries relating to the questionnaire. A researcher (PPHY) will be contactable to answer queries from those who choose to complete the questionnaire online. Participants will answer the online questionnaires at their own convenience to reduce the physical contact and time spent during their clinic visit. A weekly reminder in the form of a text message will be sent to the participants who did not complete the questionnaires for 2 weeks. Every booklet will have a bookmark inserted with a unique code and the booklets will be kept in a safe place in the clinics until collected data retrieval. In return, the participating women will receive a copy of the participant information sheet, signed consent form and a duplicate bookmark with the same code attached to their health record. The bookmark contains details on the study and the researcher contact information. Participants can request for early return of their home-based record at any time during the study period, and it will be returned to them immediately through the clinic or registered mail. At the end of the study, all home-based records will be returned to each clinic and the women will be informed for collection. They will present the duplicate bookmark for verification to collect their health records. Participants who have completed the questionnaires will receive a token of appreciation when they return to collect their antenatal home-based record.

At each participating clinic, we will extract data from the OGTT records, and only request access to the health records of the consented patients to verify necessary information including the history of miscarriage or preterm birth. Figure 2 shows the overview of the procedures in data collection.

We will also reinvite at least 50 participants from both case and control groups to complete the PBQ-14 and MPAS-19 for the 2-week intrarater test–retest reliability testing.[57] Assuming a 50% response rate, the first 100 women with and without GDM (total n=200) will be invited to participate in this reliability test. We will send the PBQ-14 and MPAS-19 online questionnaires to selected participants after 2 weeks after completing the first.

## Sample size estimation

Based on previous studies, the prevalence of GDM with adverse outcomes ranges from 5% to 27.9%.[4 11] We use GPower 3.1.9.7[60] with 0.90 power and significance at two-sided α of 0.05 to estimate the smallest difference in

**Table 1** Description of the questionnaires

| Questionnaire | Description | Score range |
|---|---|---|
| The WHO Quality of Life: Brief Version (WHOQOL)-BREF Questionnaire[46] | The WHOQOL-BREF measure is an abbreviated 26-item version of the WHOQOL-100 questionnaire and measures 4 domains of quality of life (QoL): physical (7 items), psychological (6 items), social relationship (3 items) and environment (8 items) domains and 2 additional global items focusing on overall QoL. | Four types of 5-point Likert interval scale are used, inquiring 'how much', 'how often', 'how completely', 'how satisfied' or 'how good' the respondent felt in the past 4 weeks, with different response scale distributed across the domains. Three negatively scored items are reversed scored (3, 4 and 26) and scores are summed up for each domain. Domain scores are computed by taking the mean of the scores and multiplied by 4 (and ranged from 4 to 20) to allow for direct comparison with the WHOQOL-100 scores. Higher domain scores indicate higher QoL. Malay version of this questionnaire showed high internal consistency with Cronbach's alpha ranging from 0.82 to 0.89, which is comparable to the English-language version[66] Interclass Correlation Coefficient ranged from 0.58 to 0.69 across domains, indicating good testretest reliability.[66] |
| The 14-item Postpartum Bonding Questionnaire (PBQ-14)[47 48] | The PBQ measure will be used to assess the motherinfant relationship during the postpartum period, with a total of 14 items which are rated on a six-point Likert scale from 0 (always) to 5 (never) on four subscales indicating impaired bonding, rejection and anger, anxiety about care and the risk of abuse. | When the statement is reflecting negative emotion, the scoring is reversed. The summed total score ranges from 0 to 70, with low scores indicating good bonding. The PBQ has acceptable reliability and validity and as for its utility specifically in Asian countries, the measure has been previously tested and demonstrated high sensitivity of 83% and specificity of 96%.[47 67] |
| The 19-item Maternal Postnatal Attachment Scale (MPAS-19)[49] | MPAS is a 19-item self-report questionnaire designed to assess maternal emotional response towards her infant during the first year of life. There are three dimensions:<br>1. Quality of postnatal attachment (quality of the maternal feelings towards the infant as well as maternal confidence and satisfaction in being a mother);<br>2. Absence of hostility (lack of resentment and negative feelings towards the infant) and 3) Pleasure in interaction (desire for proximity and interaction with the infant). Responses are scored on 1 (low attachment) to 5 (high attachment).[49] | The three dimensions are considered to be independent but they can be combined to obtain a global attachment score (Total postnatal attachment). The scores on the 'Quality' subscale range from 9 to 45, while the scores on the 'Pleasure in interaction' and 'Absence of hostility' subscales range from 5 to 25. Scores on the global attachment scale range from 19 to 95. Higher scores are generally indicative of stronger attachment but a specific cut-off is not provided.[49] |
| 7-item Generalised Anxiety Disorder (GAD 7)[50] | The GAD-7 is a 7-item questionnaire measuring the perceived frequency of generalised anxiety symptoms in the past 2 weeks. The items assess the most prominent diagnostic features of GAD.[68] The items include nervousness, excessive worry, and inability to stop worrying. | Restlessness, easy irritation, difficulty relaxing and fear of something awful happening on response categories 'not at all', 'several days', 'more than half the days' and 'nearly every day' scored 0, 1, 2 and 3, respectively. The summed total score ranges from 0 to 21, with higher scores indicating more severe symptoms of anxiety. The Malay version of this questionnaire was found to be valid and reliable measure in women in Malaysia, with high sensitivity of 76% and a specificity of 94%.[69] |

Continued

**Table 1** Continued

| Questionnaire | Description | Score range |
|---|---|---|
| Patient Health Questionnaire (PHQ-9)[51] | Nine items refer to symptoms experienced by patients during the 2 weeks prior to answering the questionnaire in making diagnosis and assessing severity of depression. | Scores range from 0 to 27, as each of the nine items is scored from 0 (not at all) to 3 (nearly every day). PHQ-9 scores of 5, 10, 15 and 20 represents mild, moderate, moderately severe and severe depression, respectively. This questionnaire was found to be a valid and reliable instrument to measure depression, with high sensitivity 87% and specificity of 82% in Malaysia.[70] Good internal reliability with a Cronbach's alpha of 0.67 and testretest reliability of 0.73 were also demonstrated in this population.[71] |
| The Perceived Stress Scale (PSS)[52] | The PSS measure has 10 items on a 5-point Likert scale ranging from 0 (never) to 4 (very often), assessing the perceived stress levels in the past 4 weeks. | The total score is calculated by reversing the responses for the four positively stated items (4, 5, 7 and 8) and then summing across all 10 items. The total score can range from 0 to 56, with higher score representing greater perceived levels of stress. This scale was previously used in Malaysian diabetic patients,[72] medical students,[73] working population[74] and female prisoners[75] in Malaysia and showed comparable psychometric properties to the original English version, with Cronbach's alpha from 0.63 to 0.85 and high testretest reliability of r=0.72. |

maternal or neonatal complication rate between GDM with optimal glycaemic control and suboptimal control to be at 15%, the required sample size is 312 (104 cases and 208 control). For another estimation, the required sample size to estimate 10% (as the lowest possible proportion among all the risk factors) either the history of PCOS with the reported prevalence rate 12.6%[30] or multiple pregnancies with reported incidence of 12%–30% among GDM women[32 33] with the power 0.90 and α 0.05 at two tails is 263. Taking into consideration about 10% of incomplete or missing data in the home-based records, the sample size needed is 292 (263/0.90) cases and 584 (526/0.90) control at the ratio of 1:2. Knowing the average number of pregnancies at each of the six participating clinics (200–300 cases and 1500–3000 controls per year) and the required 1:10 ratio of independent variables to numbers of dependent variables to run a multiple logistic regression with 14 independent variables on GDM with any maternal or neonatal complication rate of 40%,[61] the study is deemed to be feasible. Therefore, we will continue to collect the home-based records from the postpartum women until at least 876 records have been collected.

### Data analysis plan

All data analyses will be conducted using SPSS V.26.0 (IBM, Chicago, Illinois). Data entered will be cleaned and checked for the missing, extreme and suspicious values. These may be verified with the respondents or omitted as missing values. Once the missing data are determined to be missing at random, multiple imputations with 10 runs may be conducted to replace the missing data. The complete case analysis will be conducted if the sample size achieves 789 at a minimum.

We will use a descriptive analysis to summarise the sociodemographic data and clinical variables according to the diagnosis of GDM during the first and second trimester at 24th to 28th weeks of gestation. All glycaemic biomarkers including blood sugar profiles will be reported in the trimesters according to the outcomes (normal vs adverse). Comparisons of mean levels for continuous variables will be analysed using Student's t test and $\chi^2$ test for categorical variables. The equivalent non-parametric tests will be used for data with non-normal distribution. We will report the proportion of postnatal women who had completed OGTT in the first trimester as well as those who completed OGTT in the second trimester, and among each of this group the proportions offered OGTT twice or more (when they need to repeat the test).

To achieve the first objective, we will calculate the proportion of risk factors that are identified according to the trimester when GDM diagnosis is made. We will compare the risk factors profile of women completed OGTT and diagnosed GDM at first trimester to those who have undergone the OGTT but without GDM. A similar analysis will be conducted to compare the risk factor profiles of women with and without GDM who were offered OGTT during the second trimester and not in the first trimester. We will report the specificity, sensitivity,

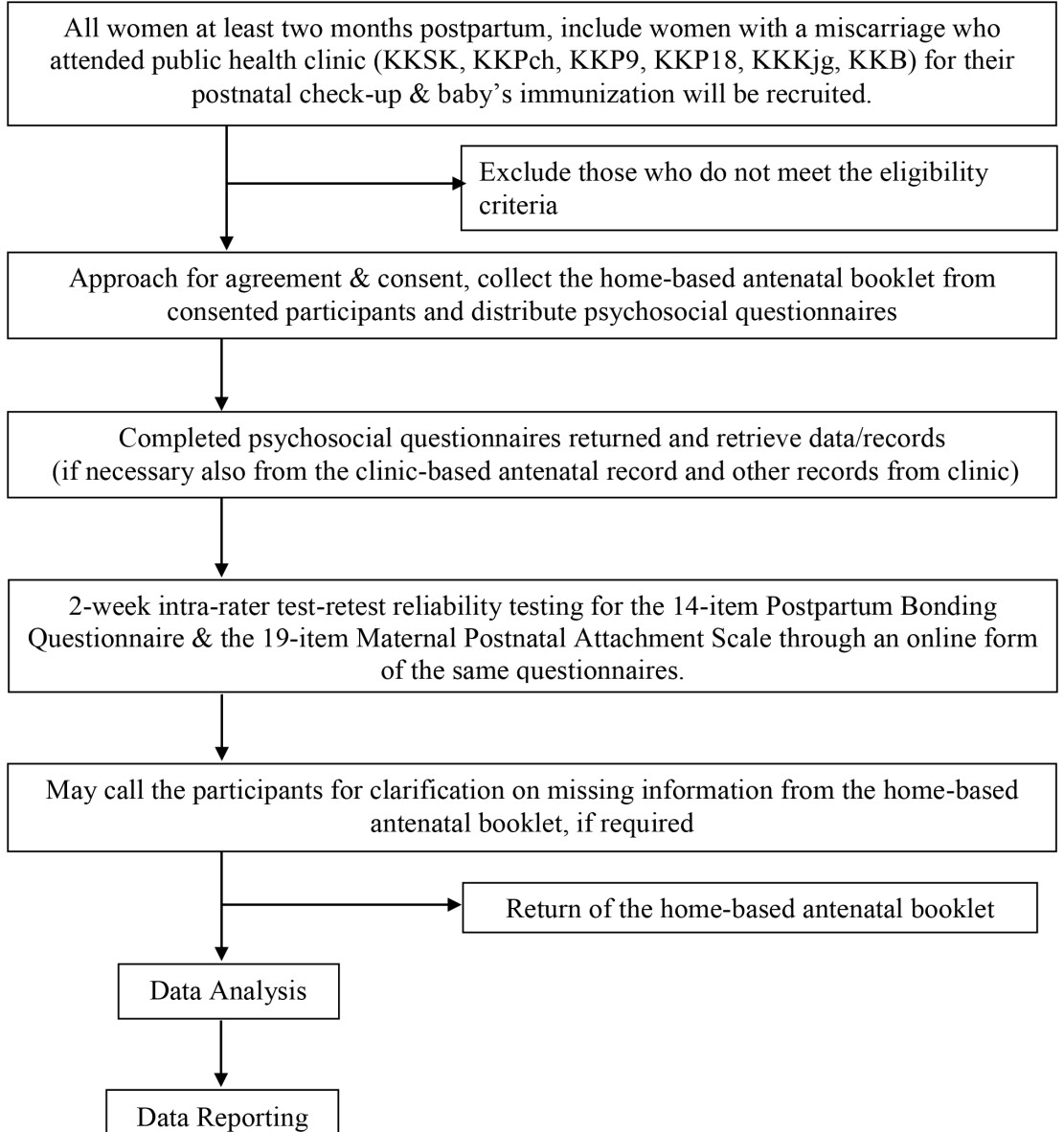

**Figure 2** Overview of the procedure in data collection.

positive and negative predictive values of the CPG-based risk factors and to examine the novel risk factors, separately and combined, for a GDM diagnosis in the first and second trimester, respectively. Additionally, we will compare the risk factors profile of women diagnosed with GDM at the first and second trimester. We will model the risk factor profiles that best predict the GDM diagnosis at the first and second trimester, respectively, using multiple logistic regression. The discrimination ability of the multiple logistic model consisting of the risk factors for GDM will be estimated with the area under the receiver operating characteristics curve with 95% CI.

We will report the comparative incidences of the maternal health outcomes during pregnancy and labour, and the comparative prevalence of the psychosocial outcomes in the postpartum period between the women with GDM and without GDM (second objective). Similarly, we will report the comparative incidences of the

neonatal outcomes between the women with GDM and without GDM (third objective).

The fourth objective can be achieved by comparing the adjusted $R^2$ values of multivariable logistic regression models consisting of the CPG-based risk factors and models with the added new potential risk factors for GDM diagnosis, including the documented glycaemic biomarkers on the outcome of maternal complications as a whole or by each complication. If sample size allows, this analysis will be conducted separately for GDM diagnosed at first and second trimester to examine its effect on the maternal complications.

Analysis step to measure the fifth objective is similar to the fourth objective by comparing the adjusted $R^2$ values of multivariable logistic regression models consisting of the CPG-based risk factors and models with the added new potential risk factors for GDM diagnosis, including

documented glycaemic biomarkers on neonatal health outcomes.

The current risk factors for GDM will be assessed univariably and multivariably of their effect on the diagnosis of GDM at first and second trimester ($24^{th}$ to $28^{th}$ weeks) using the $R^2$ and adjusted $R^2$. The factors from the demographic and clinical variables on GDM diagnosis will be estimated in univariable logistic regression analyses. Any of this factor with a *P*-value<0.20 from will be included in the multiple logistics regression analysis.[62] The analyses may be conducted by blocks of risk factors such as sociodemographic information (age, BMI and smoking), family history and past medical history, current antenatal medical problems (glycosuria, obstetric medical conditions and weight gain) if sample size is less than desirable. Multicollinearity between any independent variables will be checked using correlation matrix and standard errors (SE) of each variable.[63] Any two variables correlated >0.9 or/and the variable has a SE >5.0 will indicate the presence of multicolinearity. In the presence of multicollinearity, the variable with largest SE and less critical or essential from clinical perspectives will be excluded. This process will continue until the magnitude of the SEs for all the variables hover around 0.001–5.0. All final models, Q-Q plots for normality, the residual plots for linearity and homogeneity assumptions and model fitting will be checked. Same statistical strategy may be conducted to model the independent predictors on the maternal and neonatal outcomes. The maternal outcomes (abnormal gestational weight gain, pre-eclampsia, polyhydramnios, abnormal modes of delivery, gestational age at birth either pre-term or post-date birth and complication in labour includes postpartum haemorrhage and perineal tear) will be combined as a whole or separately if sample size for any of the outcome allows, and the maternal psychosocial well-being measures will be analysed separately. Similarly, if sample size allows, we will analyse for the neonatal outcomes (birth weight, abnormal gestational weight at birth, poor Apgar score, congenital anomaly, congenital hypothyroidism, neonatal death, stillbirth, hypoglycaemia, and hyperbilirubinaemia). Confounding factors will be assessed in multiple logistic regression modelling on maternal and neonatal outcomes, and maternal psychosocial well-being measures. A confounding factor is present when it changes the odd ratio of GDM on the outcome by a magnitude of > 10%.[64] This will be done to verify the variables included in the final models of the maternal and neonatal outcomes, and maternal psychosocial well-being measures.

## Strengths and Limitations

This retrospective cohort study incorporates a nested case-control design, providing an opportunity to confirm and explore the conventional CPG-based and potential novel risk factors to better predict the diagnosis of GDM, either at the first or/and second trimester. This is one of the core outcomes to be included in GDM prevention and treatment research.[65] Additionally, we will also examine

multiple pregnancies on diagnosis of GDM, which was often excluded from earlier studies. The risk profiles that best predict the diagnosis of GDM will be modelled and examined for their impacts on the maternal and neonatal health outcomes. This evidence is imperative to improve the identification of at-risk women and earlier treatment for GDM. This study would help healthcare practitioners and women with GDM to better understand the effect of both the currently 'recommended' and potentially 'new' important risk factors of GDM, patterns of glycaemic control and their association with health outcomes in both the women and neonates. This is potentially impactful on the decision rule of the existing practice. As this study will be assessing the level of OGTT completion, it will determine whether a delay in the completion of OGTT is one of the possible causes of a delayed GDM diagnosis and treatment, and the risk profiles of women and their association with any maternal and neonatal health outcomes will be identified. Additionally, the effect of each individual risk factor and as a whole on GDM, glycaemic patterns during pregnancy, and the maternal and neonatal health outcomes including psychosocial well-being will be quantified. Furthermore, this study would also validate locally and culturally adapted 14-item PBQ and the 19-item MPAS.

This study faces a few limitations. The incidence of some of the CPG-based and potential novel risk factors may be low and insufficient in numbers for the multivariable regression analysis. To overcome this, the study has included all the adjacent six public health clinics. Consequently, this has also increased the demands of training and supervision of research assistants, travelling time and coordinating effort. This will be taken care of by having a written study manual, a site visit to the participating clinics before the start of recruitment, and regular contact with all the research assistants until the end of recruitment. In the case of a smaller than expected sample size being recruited, multivariable regression analysis by separate blocks of risk factors will be conducted. Second, the six public health clinics are situated in the urban areas of Selangor and Putrajaya, thus the study participants may not be representative of the larger Malaysian population. Owing to the COVID-19 pandemic, the effects of the aftermath may delay the data collection processes and cause a reduction in the number of pregnant women visiting the clinics in-person. There may also be a decrease in pregnancy in the past year which would lower the number of postpartum women participants with a history of GDM this year. Another limitation may be the quality and accuracy of the data in the antenatal health records. However, we believe this problem is minimal with the long use over decades of the same antenatal records by all the healthcare providers in the public health clinics. The booklet is well-structured with dedicated spaces for the variables to be investigated in this study. We plan clarification and verification strategies to confirm the nature of the risk factors when present with doubts by contacting the participant, checking the clinic-based or/and hospital-based records.

## Patient and public involvement

This is one of non-experimental studies in the MYGOD-DESS Project (https://rb.gy/ccztw5) where women with GDM during pregnancy and in postpartum periods will be interviewed on important barriers and facilitators of self-care. Women with a history of GDM are involved in the face and content validity testing of the PBQ-14 and MPAS-19 questionnaires. Their opinions on the study design, conduct, reporting and dissemination of results will be sought at appropriate time during the study.

## ETHICS AND DISSEMINATION
### Risk-benefit assessment

Participants are not subjected to any medications or treatments during the study period. No rescue medication or procedures will be involved. Since this is not an interventional study, there is therefore no direct health risk and no side effects for the participant. There might be a potential risk of fatigue on completion of the questionnaires. No direct benefit to participants. A small token of appreciation will be given to all participants. Their participation will provide data that aims to improve and increase understanding in the research topic that may contribute to future protocol, guidelines or policymaking.

### Ethical consideration

The study will be conducted in compliance with the ethical principles outlined in the Declaration of Helsinki and Malaysian Good CPG. This study obtained approval from the MREC ethics committee. We will clearly explain, state the purpose of this study, and to obtain written consent in Malay or English from all study participants before data collection. Participation is voluntary. They have the right to withdraw at any stage of the research without giving any reason. Should there be any further amendments to the protocol, other than administrative ones, further approval from the MREC ethics committee will be obtained. Any revisions of documents and amendment to the protocol originally submitted for review, unexpected events during the study period, and new information that may adversely affect the safety of participants and publication will duly be informed to the ethics committee.

### Privacy and confidentiality

Only researchers of this study have access to the participants' data and will be handled diligently only for the purpose of this study. Participant identity will not be revealed as there will be no referencing of participants by name on presenting the result. The identification number will be used on subject data sheets. All information in this study is confidential. Data from this study will be entered and saved on a dedicated computer that is password protected. On completion of the study, softcopy data in the computer will be copied to a password protected pen drive and the data in the computer will be erased. Any hardcopy (including the consent form) and the pen drive

will be kept in the Principal Investigator's locked office at UPM and maintained for a minimum of twenty years after the completion of the study. The collected data will be destroyed after that period of storage. Subjects will not be allowed to view their personal study data as the data will be consolidated into a database. The participants can write to the investigators to request access to the study findings.

### Publication policy

Participants' personal information will not be disclosed, thus will not be identified when the findings of this research are published and presented.

### Dissemination plan

Research findings will be published in scientific journals, may be presented in scientific conferences, and will be reported and shared with the local health stakeholders.

**Author affiliations**
[1]Department of Family Medicine, Faculty of Medicine and Health Sciences, Universiti Putra Malaysia, Serdang, Selangor, Malaysia
[2]Faculty of Epidemiology and Population Health, London School of Hygiene and Tropical Medicine, London, UK
[3]Department of Psychological Medicine, King's College London Institute of Psychiatry Psychology and Neuroscience, London, UK
[4]Department of Obstetrics and Gynaecology, Faculty of Medicine and Health Sciences, Universiti Putra Malaysia, Serdang, Selangor, Malaysia
[5]Department of Pathology, Faculty of Medicine and Health Sciences, Universiti Putra Malaysia, Serdang, Selangor, Malaysia
[6]Division of Care in Long-term Conditions, King's College London, London, UK
[7]Clinical Research Centre (CRC), Hospital Putrajaya Malaysia, Putrajaya, Malaysia
[8]Klinik Kesihatan Seri Kembangan, Ministry of Health Malaysia, Seri Kembangan, Selangor, Malaysia
[9]Klinik Kesihatan Puchong, Ministry of Health Malaysia, Puchong, Selangor, Malaysia
[10]Klinik Kesihatan Putrajaya Presint 9, Ministry of Health Malaysia, Putrajaya, Malaysia
[11]Klinik Kesihatan Putrajaya Presint 18, Ministry of Health Malaysia, Putrajaya, Malaysia
[12]Klinik Kesihatan Kajang, Ministry of Health Malaysia, Kajang, Selangor, Malaysia
[13]Klinik Kesihatan Bangi, Ministry of Health Malaysia, Bangi, Selangor, Malaysia
[14]Clinical Research Unit, Hospital Pengajar Universiti Putra Malaysia, Serdang, Selangor, Malaysia

**Contributors** BHC, NIB, INS and PPHY contributed to the study planning and design of the study. The study design was further developed by IPN, VR, MB and BHC. PPHY drafted the manuscript with assistant from BHC and also input from various stages from NMN, ZAS, HH, FP, RZ, SRMA, NBIB, AF and KI. All authors critically reviewed the manuscript and approved the final version as submitted.

**Funding** This study is funded by MYPAIR Grant UK-Malaysia: Joint Partnership Call on Non-Communicable Diseases (Malaysia: JPT.S(BPKI)2000/011/06/05 (27); UK: MR/T018240/1).

**Competing interests** None declared.

**Patient and public involvement** Patients and/or the public were involved in the design, or conduct, or reporting, or dissemination plans of this research. Refer to the Methods section for further details.

**Patient consent for publication** Not applicable.

**Ethics approval** This study was approved by Malaysia Research and Ethics Committee (MREC), Ministry of Health Malaysia (MOH) with the reference number NMRR-20-3000-57095 (IIR). Participants gave informed consent to participate in the study before taking part.

**Provenance and peer review** Not commissioned; externally peer reviewed.

Data sharing will be considered on a case-by-case basis upon assessment of the proposed study protocol.

**ORCID iDs**
Pamela Phui Har Yap http://orcid.org/0000-0002-5763-4114
Angus Forbes http://orcid.org/0000-0003-3331-755X
Boon How Chew http://orcid.org/0000-0002-8627-6248

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
