## [Reviewer comments · BMJ Open]

ARTICLE DETAILS

TITLE (PROVISIONAL)	Study protocol on Risk factors for the diagnosis of gestational diabetes mellitus in different trimesters and their relation to maternal and neonatal outcomes (GDM-RIDMAN)
AUTHORS	Yap, Pamela; Papachristou Nadal, Iliatha; Rysinova, Veronika; Basri, Nurul Iftida; Samsudin, Intan; Forbes, Angus; Noor, Nurain; Supian, Ziti; Hassan, Haslinda; Paimin, Fuziah; Zakaria, Rozita; Mohamed Alias, Siti; Ismail Bukhary, Norizzati; Benton, Madeleine; Ismail, Khalida; Chew, Boon How

VERSION 1 – REVIEW

REVIEWER	Baradaran, Hamid Reza Iran University of Medical Sciences, Center for Educational Research in Medical Sciences (CERMS)
REVIEW RETURNED	29-Jun-2021

GENERAL COMMENTS	Thank the authors for an interesting study "Risk factors for the diagnosis of gestational diabetes mellitus in different trimesters and their relation to maternal and neonatal outcomes (GDM-RIDMAN): a retrospective cohort and nested case-control study" Issues to be addressed are as follows: - There is no need to mention the term "a retrospective cohort" in the title and other parts of the manuscript.- Since the authors plan to examine the diagnostic accuracy of the latest Malaysian CPG-based risk factors, they need, it is necessary to estimate the ROC curve and AUC with 95% confidence interval.- It is not clear whether logistic regression model will be conditional or classic given the nature of the nested case-control study and type of matching.- Given the assessment of a considerable number of independent variables or predictors, it is better to consider a relatively larger sample size (due to reduced study power and predictive power of model and sparse data and reduced precision of the estimates).
--

REVIEWER	Nsiah, Paul University of Cape Coast, Chemical Pathology
REVIEW RETURNED	27-Jul-2021

GENERAL COMMENTS	Given the various controversies about GDM, from classification and diagnosis to treatment, this will be a good study.
---

REVIEWER	Li, Yuanyuan Tongji Medical College
-----------------	--

REVIEW RETURNED	04-Jan-2022
-------------

GENERAL COMMENTS	This article considers the deficiencies and controversies of current GDM diagnosis from a new perspective, introduces some new variables as risk factors for screening GDM, and explores the relationship between GDM and adverse pregnancy outcomes, including psychological outcomes. It is an active attempt to improve the current GDM screening and diagnostic standards. But there are still some issues that need to be pointed out:  1. The data of the study comes from six public health clinics. How to reduce the heterogeneity of GDM diagnosis and questionnaire surveys from different clinic sources? 2. On page 10, please explain in detail what criteria the study participants of the control group were selected according to. 3. On page 13, the article mentioned that if the sample size allows, the maternal outcome will be included in the model as a whole. Please explain this part in detail. 4. On page 13, how to solve the collinearity problem in multiple logistic regression? Please give a more detailed explanation. 5. How to determine confounding factors when constructing the multiple logistic regression model? 6. The incidence of some GDM risk factors or adverse pregnancy outcomes is low, and there may not be enough data for multivariate regression analysis. How will this study solve this problem? 7. The grammatical accuracy and fluency of the article needs to be improved.
--

VERSION 1 – AUTHOR RESPONSE

Response to reviewers' comments

- Please clearly state that this is a protocol in the Title.

We have revised the title to “Study protocol on Risk factors for the diagnosis of gestational diabetes mellitus in different trimesters and their relation to maternal and neonatal outcomes (GDM-RIDMAN)”

- Please reformat the abstract so that it follows the structured abstract recommended in the journal's instructions for authors for research articles. See: <https://bmjopen.bmj.com/pages/authors/#research>
Abstract reformatted, please refer page 3

- Please revise the Strengths and Limitations section of your manuscript (after the Abstract). This section should contain up to five short bullet points, no longer than one sentence each, that relate specifically to the methods. Please do not discuss the potential results or implications of this study in this section.

Strengths and Limitations revised, please refer page 4

- Please reformat the main text so that it follows the structure recommended in the journal's instructions for authors for study protocols, for example the main text of your manuscript should contain an Ethics and Dissemination section. See: <https://bmjopen.bmj.com/pages/authors/#protocol>
We reformatted and revised the manuscript, Ethics and Dissemination section has been included while discussion is under Strength and Limitation subsection under Methods and analysis.

Formatting Amendments (where applicable): Done.

Reviewer: 1

Dr. Hamid Reza Baradaran, Iran University of Medical Sciences

Comments to the Author:

Thank the authors for an interesting study “Risk factors for the diagnosis of gestational diabetes mellitus in different trimesters and their relation to maternal and neonatal outcomes (GDM-RIDMAN): a retrospective cohort and nested case-control study” Issues to be addressed are as follows:

1. There is no need to mention the term "a retrospective cohort" in the title and other parts of the manuscript.

We have amended the title to “Study Protocol on Risk factors for the diagnosis of gestational diabetes mellitus in different trimesters and their relation to maternal and neonatal outcomes (GDM-RIDMAN).”

2. Since the authors plan to examine the diagnostic accuracy of the latest Malaysian CPG-based risk factors, they need, it is necessary to estimate the ROC curve and AUC with 95% confidence interval. Thank you for your suggestion. We will do so with the final multiple logistic model that comprises significant risk factors in the study. We have added this to the revised manuscript at page 12, para 2, last 3 lines: “The discrimination ability of the multiple logistic model consisting of the risk factors for GDM will be estimated with the area under the receiver operating characteristics curve with 95% confidence interval.”

3. It is not clear whether logistic regression model will be conditional or classic given the nature of the nested case-control study and type of matching.

The study will be using the classical instead of Bayesian logistic regression model conditional on significant covariates identified through preceding univariable logistic regression analyses as described on page 13, para 3. We apply little matching in the sampling except on the timing of GDM to avoid overmatching and dilution of effect of the risk factors, and this is in line to the principles of a case-control study (Grobbee DE, Hoes AW: Clinical Epidemiology: Principles, Methods, and Applications for Clinical Research: Jones & Bartlett Learning; 2014).

4. Given the assessment of a considerable number of independent variables or predictors, it is better to consider a relatively larger sample size (due to reduced study power and predictive power of model and sparse data and reduced precision of the estimates).

We agree with the thought. Towards that, we have conducted the sample size estimation and feasibility check with the participating clinics, please see page 11-12. We come to 876 to be collected assuming 10% incomplete rate or 789 complete cases.

Reviewer: 2

Dr. Paul Nsiah, University of Cape Coast

Comments to the Author:

Given the various controversies about GDM, from classification and diagnosis to treatment, this will be a good study.

Thank you for your affirmation.

Reviewer: 3

Dr. Yuanyuan Li, Tongji Medical College

Comments to the Author:

This article considers the deficiencies and controversies of current GDM diagnosis from a new perspective, introduces some new variables as risk factors for screening GDM, and explores the relationship between GDM and adverse pregnancy outcomes, including psychological outcomes. It is

an active attempt to improve the current GDM screening and diagnostic standards. But there are still some issues that need to be pointed out:

1. The data of the study comes from six public health clinics. How to reduce the heterogeneity of GDM diagnosis and questionnaire surveys from different clinic sources?

GDM is diagnosed based on the same Malaysian 2017 CPG on Management of Diabetes in Pregnancy, when either one of the OGTT results, FPG ≥ 5.1 mmol/L or 2-HPP ≥ 7.8 mmol/L (please see page 9, first 3 lines). Questionnaires WHOQOL-Bref, GAD-7, PHQ-9, and PSS-10 are locally validated (Table 3), while the 14-item Postpartum Bonding Questionnaire (PBQ-14) and 19-item Maternal Postnatal Attachment Scale (MPAS-19) will be validated in the study.

The patient population in the six participating public health clinics are expected to provide a study samples that is representative of the Malaysian population from the important demographic characteristics of age, gender and ethnicity. For this reason, the analysis plan is taking the approach of analysing the whole sample for the main study objectives. For the objectives on maternal and neonatal outcomes, we will assess whether there would be any significant differences in the incidences in the six clinics. If there is a substantial and significant difference, we will adjust for it using a mixed-effect modelling if feasible (may not be able to robustly with < 10 sites) or included the clinic as a separate covariate.

2. On page 10, please explain in detail what criteria the study participants of the control group were selected according to.

We have revised the sentences, refer to page 10 (sampling process), "All women with GDM will be the case. Two control will be sampled for each case. Women in control group were selected based on the eligibility criteria from the same six health clinics who had undergone OGTT but without a diagnosis of GDM, at about the same period of gestation at the diagnosis of GDM to the case"

3. On page 13, the article mentioned that if the sample size allows, the maternal outcome will be included in the model as a whole. Please explain this part in detail.

We have improved the sentences and added details in the revised manuscript (refer to page 13): "The maternal outcomes (abnormal gestational weight gain, pre-eclampsia, polyhydramnios, abnormal modes of delivery, gestational age at birth either pre-term or post-date birth and complication in labour includes postpartum haemorrhage and perineal tear) will be combined as a whole or separately if sample size for any of the outcome allows, and the maternal psychosocial wellbeing measures will be analysed separately."

We have mentioned in the text earlier at page 11, last 3 lines that "... 1:10 ratio of independent variables to numbers of dependent variable to run a multiple logistic regression..."

4. On page 13, how to solve the collinearity problem in multiple logistic regression? Please give a more detailed explanation.

We have revised the approach in the multicollinearity checking to "Multicollinearity between any independent variables will be checked using correlation matrix and standard errors (SE) of each variable. Any two variables correlated > 0.9 or/and the variable has a SE > 5.0 will indicate the presence of multicollinearity. In the presence of multicollinearity, the variable with largest SE and less critical or essential from clinical perspectives will be excluded. This process will continue until the magnitude of the SEs for all the variables hover around 0.001 – 5.0." (please see page 13)

5. How to determine confounding factors when constructing the multiple logistic regression model?

The study will select the potential predictors in univariable logistic regression analyses with a P-value < 0.20 (see page 13, para 3, line 3-5) for the GDM diagnosis, and there is no confounding factor by definition in a diagnostic study because they are predictors or determinants.

Confounding factors will be assessed in multiple logistic regression modelling on maternal and neonatal outcomes, and maternal psychosocial wellbeing measures. A confounding factor is present when it changes the odd ratio of the main factor which is GDM on the outcome by a magnitude of > 10%. We have added this as another approach to verify the final models of the maternal and neonatal outcomes, and maternal psychosocial wellbeing measures (see page 13 last 3 lines).

6. The incidence of some GDM risk factors or adverse pregnancy outcomes is low, and there may not be enough data for multivariate regression analysis. How will this study solve this problem? If the event is very low (for example < 10) for any single outcome, the study will not be able to model a multivariable analysis. For any outcome that cannot accommodate all the independent variables because of insufficient event rate, the study will either combine them or conduct the modelling by blocks of risk factors such as sociodemography (age, BMI and smoking) as the first block, family history and past medical history as another, current antenatal medical problems (glycosuria, obstetric medical conditions and weight gain) as another.

7. The grammatical accuracy and fluency of the article needs to be improved. We have the English native speaker authors reading the manuscript through again.

VERSION 2 – REVIEW

REVIEWER	Li, Yuanyuan Tongji Medical College
REVIEW RETURNED	28-Feb-2022
GENERAL COMMENTS	The manuscript has been much improved.